# Gas Chromatography–Mass Spectrometry Profiling of Volatile Compounds Reveals Metabolic Changes in a Non-Aflatoxigenic *Aspergillus flavus* Induced by 5-Azacytidine

**DOI:** 10.3390/toxins12010057

**Published:** 2020-01-19

**Authors:** Fengqin Song, Qingru Geng, Xuewei Wang, Xiaoqing Gao, Xiaona He, Wei Zhao, Huahui Lan, Jun Tian, Kunlong Yang, Shihua Wang

**Affiliations:** 1School of Life Science, Jiangsu Normal University, Xuzhou 221116, China; songfengqin@jsnu.edu.cn (F.S.); qingrugeng@jsnu.edu.cn (Q.G.); wangxuewei@jsnu.edu.cn (X.W.); gaoxiaoqing@jsnu.edu.cn (X.G.); hexiaona@jsnu.edu.cn (X.H.); zhaowei00@jsnu.edu.cn (W.Z.); 2Key Laboratory of Pathogenic Fungi and Mycotoxins of Fujian Province, Key Laboratory of Biopesticide and Chemical Biology of Education Ministry, and School of Life Sciences, Fujian Agriculture and Forestry University, Fuzhou 350002, China; huahuilan90mail@fafu.edu.cn

**Keywords:** *Aspergillus flavus*, DNA methylation, aflatoxin, volatiles

## Abstract

*Aspergillus flavus* is one of the most opportunistic pathogens invading many important oilseed crops and foodstuffs with such toxic secondary metabolites as aflatoxin (AF) and Cyclopiazonic acid. We previously used the DNA methylation inhibitor 5-azacytidine to treat with an AF-producing *A. flavus* A133 strain, and isolated a mutant (NT) of *A. flavus*, which displayed impaired abilities of AF biosynthesis and fungal development. In this study, gas chromatography–mass spectrometry (GC-MS) analysis was used to reveal the metabolic changes between these two strains. A total of 1181 volatiles were identified in these two strains, among which 490 volatiles were found in these two strains in vitro and 332 volatiles were found in vivo. The NT mutant was found to produce decreasing volatile compounds, among which most of the fatty acid-derived volatiles were significantly downregulated in the NT mutant compared to the A133 strain, which are important precursors for AF biosynthesis. Two antioxidants and most of the amino acids derived volatiles were found significantly upregulated in the NT mutant. Overall, our results reveal the difference of metabolic profiles in two different *A. flavus* isolates, which may provide valuable information for controlling infections of this fungal pathogen.

## 1. Introduction

*Aspergillus flavus* is a soil-born crossover pathogen that could opportunistically invade many important oilseed crops and immunosuppressed patients [1,2]. Importantly, the toxigenic *A. flavus* produces various harmful secondary metabolites (SMs) when spoiling foodstuffs, among which aflatoxins (AFs) are the most notorious and carcinogenic mycotoxin [3,4]. Foodstuff contaminated with a low dose of AFs that is ingested chronically by animal or human could pose liver cancer, while ingestion of a high dose of AFs contaminating foodstuffs is poisonous and even fatal [5,6,7,8]. Adding to AFs, cyclopiazonic acid (CPA), aflatrem, and kojic acid (KA) were also identified in *A. flavus* [9,10]. These SMs produced by *A. flavus* are small bioactive molecules that exhibit harmful for animals and humans [11]. The whole genome of *A. flavus* has been sequenced [12,13], which reveals 56 secondary metabolite clusters that are regulated by different environmental factors [14]. Although each SM cluster can potentially produce specific metabolites, only the molecular structures of AFs, CPA, aflatrem, and KA are identified in this fungus [15].

SMs are rarely found to be involved in fungal development or reproduction, but they do play critical roles in fungal ecology in response to environmental factors [16]. In *Aspergillus*, although some SMs exhibit harmful properties (like AFs and CPA), some other SMs like lovastatin also display beneficial properties for humans [9,17]. The SM genes, which are necessary for synthesizing a specific compound, are usually clustered in the genome. In filamentous fungi, most of the SM clusters are located proximately to the telomere or heterochromatin, where they are subject to specific silencing mechanisms [18]. The silencing mechanisms in these chromatin regions are associated with histone acetylation and DNA methylation [19]. In filamentous fungi, the inhibitors of histone deacetylases (HDACs) and DNA methyltransferases (DNMTs) have been utilized to inhibit gene silencing in the genome and activate the expression of SM gene clusters [19,20,21]. The inhibitors of the second class HDACs, such as trichostatin A (TSA), SAHA (vorinostat), and suberoyl bis-hydroxamic acid (SBHA) [19,21,22], have been utilized to successfully identify new SMs in fungi. TSA was the first HDAC inhibitor used to increase SM production in *Aspergillus* [21]. In *A. nidulans*, the treatment of 5-methylmellein, an inhibitor of HDAC SirA produced by *Didymobotryum rigidum*, promoted SM profiles [23]. 

The DNA methyltransferases inhibitors 5-azacytidine and RG-108 have also been used to identify novel metabolites in fungi [20,24]. Although the *Aspergillus* species exhibit low concentration of DNA methylation in their genomes, our former study has demonstrated that the use of 5-azacytidine in *A. flavus* resulted in a dramatic reduction of AF production and concurrent developmental defects [25]. Additionally, a non-aflatoxigenic mutant (NT) of *A. flavus* induced by 5-azacytidine was also isolated, which displayed impaired abilities of AF biosynthesis and fungal development. Fungal volatiles are important cellular metabolites, which hold a considerable portion of the microbial metabolome [17]. In this study, we performed comparative volatile profiling to analyze the information on intracellular metabolism in an aflatoxigenic *A. flavus* A133 strain and a non-aflatoxigenic mutant (NT) induced by 5-azacytidine by gas chromatography–mass spectrometry (GC-MS). A total of 1181 volatiles were identified in A133 or NT strains, among which 490 volatiles were found in these two strains in vitro and 332 volatiles were found in vivo.

## 2. Results

### 2.1. The NT Mutant Induced by 5-Azacytidine Is Impaired in Conidiation and Mycotoxins Production

DNA methylation, an important epigenetic modification, plays a critical role in the regulation of gene transcription involving in development and secondary metabolites in many plants and fungi. Our former study has demonstrated that the treatment of 5-azacytidine, a DNA methyltransferase inhibitor, blocks AF biosynthesis and fungal development in *A. flavus* [25]. We also obtained a non-aflatoxigenic mutant (NT) of *A. flavus* by 5-azacytidine treatment, which failed to produce AF. To better understand the changes in this mutant induced by 5-azacytidine, we both examined its developmental phenotype and mycotoxins production compared to the A133 wild-type strain. As shown in Figure 1A, the NT strain produced white fluffy mycelium and failed to produce conidia pigment, while the A133 strain produced dense dark green spores. Further examination of aerial conidiophores by microscope also showed that the NT strain produced fewer conidial heads and could not form normal conidiophores compared to the A133 strain (Figure 1B). The abilities of mycotoxins production of the NT strain in a Czapek liquid medium were also determined. As shown in Figure 1C, the A133 strain produced yellow mycelial pellets and grew with adherence, while the NT strain produced white and bigger mycelial pellets and grew without adherently. We then extracted their secondary metabolites (SMs) from the cultures with chloroform, and the result showed that the extracts of the A133 strain displayed yellow in color, while the NT strain was colorless. Then the SMs extracts were separated using thin-layer chromatography (TLC), which showed that the A133 strain was able to produce aflatoxin B1 (AFB1) and AFB2 both in vivo (volatiles extracted from the mycelium of *A*. *flavus*) and in vitro (volatiles emitted from the *A*. *flavus*), while non detectable aflatoxins were found in the NT strain in vitro (Figure 1D). Intriguingly, AFB2 but not AFB1 was found in the NT strain in vivo (Figure 1D).

### 2.2. GC-MS Analysis of Volatiles Extracts

Fungal volatiles are important cellular metabolites which hold a considerable portion of the microbial metabolome. To analyze the diversity of volatile profiles within the A133 (WT) and NT strains, GC-MS analysis was performed to detect the volatiles emitted from these two strains (in vitro) or extracted from their mycelium (in vivo). A representative total ion chromatogram (TIC) was shown in Figure 2A (in vitro) and Figure 2B (in vivo), and indicates the different volatile profiles among the A133 and NT strains. The resulted GC-MS profiles were subjected to XCMS online program (https://xcmsonline.scripps.edu/) for further analysis. As shown in Figure 3A,B, the results of principal component analysis (PCA) demonstrated that NT mutants clustered separately with A133 wild-type strain, which implies that they have specific metabolic profiles under aflatoxin-producing condition in comparison to wild type (PCA 1, 90%), as well as to each other (PCA 2, 3%). Metabolite with *p* value ≤ 0.01 and fold change ≥ 1.5 in each comparison was selected as a changed metabolite (Figure 3C,D).

### 2.3. The A. flavus Non-Aflatoxigenic Mutant Produces Decreasing Volatile Compounds

The NT mutant was compared to the wild type both in vitro and in vivo within the XCMS program and visualized using volcano plots according to log_10_ of *p* value (y-axis) and log_2_ of fold change (x-axis) (Figure 4A,B). Here we identified a total of 849 and 691 volatiles in A133 or NT strains in vitro and in vivo, respectively (Figure 4C), among which 490 specific volatiles were identified in these two strains in vitro and 332 specific volatiles in vivo. Intriguingly, the number of volatile compounds observed in the NT strain (531/535) was reduced compared to the wild-type strain (567/559) in vitro and in vivo (Figure 4A,B).

### 2.4. Screening of the Changed Volatile Compounds in NT Mutant

The identified metabolites were clustered and visualized with a heat map, and the results showed that, among the 849 volatile metabolites emitted extracellularly from these two strains (in vitro), 376 compounds showed increasing expression in the A133 strain compared to the NT mutant, while 473 compounds were significantly up-regulated in the NT mutants compared to the A133 strain (Appendix A). A total of 691 volatile metabolites were identified from the mycelium of these two strains (in vivo), among which 450 compounds were significantly up-regulated in the A133 strain compared to the NT mutant, while 241 compounds were shown to be over-expressed in the NT mutants compared to the A133 strain (Appendix A). The top 30 volatile metabolites produced by the A133 strain and NT strain in vitro, and the top 20 volatile metabolites in vivo, differing according to *t*-test were also clustered and visualized with a heat map (Figure 5A,B). A total of 21 compounds were up-regulated in the A133 strain among the top 30 metabolites, and 9 compounds were found increasing expression in NT mutants (Figure 5A). These top differential expressed metabolites were classified, which included fatty, amino, organic acid, as well as the saccharides. The fatty acid-derived volatiles and most of the phenylpropanoid derivatives were significantly up-regulated in the A133 strain compared to the NT strain (Table 1 and Table 2). The A133 strain accumulated extremely higher levels of n-Hexadecanoic acid (more than 2846 fold levels vs. in the NT mutant), 9,12-Octadecadienoicacid(Z,Z)-(more than 1980 fold levels vs. in the NT mutant), Ergosterol (more than 106 fold levels vs. in the NT mutant) and Flopropione (more than 140 fold levels vs. in the NT mutant) extracellularly. Among these top metabolites, M69T60 (2,6,10,14,18,22-Tetracosahexaene,2,6,10,15, 19,23-hexamethyl-,(all-E)-) was the only triterpene identified in the A133 strain, which was detected with more than 82 fold levels in A133 vs. in the NT mutant (Table 1). Most of the differential expressed amino acids derived volatiles were detected highly expressed in the NT mutant both in vitro and in vivo with more than 66 fold levels vs in the A133 strain (Table 1 and Table 2). Among the top differential expressed metabolites extracted from the fungal mycelium (in vivo), accumulation of three phenylpropanoid derivatives (4 in total) and one saccharide derived volatile (D-Glucoside), the only saccharide derived volatile identified in these two strain in vivo, were found in the NT mutant compared to the A133 strain (Table 2). In the NT mutant, one of the phenylpropanoid derivatives, M353T55 (Phenol,4,4′-methylenebis [2,6-bis (1,1-dimethylethyl)]-) was detected with more than 9 fold levels vs. the A133 strain, which is considered as an antioxidant.

The volatile metabolites were also analyzed with the Agilent ChemStation software, which showed that a total number of 84 compounds emitted extracellularly were identified from these two *A. flavus* strains (in vitro), and a total amount of 54 compounds were identified in vivo from the mycelium of these two strains (Appendix A). The A133 strain and NT mutant shared 18 and 28 same compounds produced in vitro and in vivo, respectively. Among these identified volatile metabolites, most of the fatty acid-derived volatiles were up-regulated in the A133 strain produced in vitro (Appendix A). While most of the amino acid-derived volatiles, intriguingly, were detected increasingly in the NT mutant (Appendix A). The properties of the metabolites were identified both in the websites of PubChem database (https://pubchem.ncbi.nlm.nih.gov) and Guidechem (https://www.guidechem.com/). Here, we found that two metabolites, Phenol,2,2′-methylenebis[6-(1,1-dimethylethyl)-4-methyl-] (CAS: 000119-47-1) and Phenol,4,4′-methylenebis[2,6-bis(1,1-dimethylethyl)-] (CAS: 000118-82-1), both of which were considered as antioxidants, were found significantly upregulated in the NT mutant (Appendix A), which is consistent with the result analyzed by the XCMS program.

## 3. Discussion

*A. flavus* is a notorious crossover pathogen, disseminating in agricultural soils, that could not only spoil crop seeds and foodstuffs but also be a pathogen of immunosuppressed patients [2]. The contamination of aflatoxins (AFs) and other mycotoxins produced by this fungus on foodstuff or crop seeds has been the cause of significant concern within the food industry and farmers [26,27]. This fungus is hard to eliminate from the environment as the formation of resistant overwintering structures, sclerotia. Our former work indicated that treatment with the DNA methylation inhibitor 5-azacytidine (5-AC) inhibited the reproduction of *A. flavus* and blocked AFs biosynthesis [25]. In our previous study, the 5-AC was further used to successfully isolate an *A. flavus* mutant [25], which displayed impaired abilities of AF biosynthesis and fungal development. Here, we performed a GC-MS analysis to reveal the metabolic changes in this mutant. A total number of 1181 volatiles were identified in the NT mutant and wild-type strain, among which 490 emitted compounds were identified in these two strains, and 332 intracellular compounds were found in these two strains intracellularly. Intriguingly, the decreasing volatile compounds were observed in the NT mutant compared to the wild-type strain, which could result in such physiological changes as reducing AF production and asporulate phenotype in the NT mutant.

In filamentous fungi, DNA methyltransferases inhibitors like 5-azacytidine have been used to identify novel metabolites re-expressing the methylation-silenced SM genes [20,24]. Although the existence of DNA methylation in *Aspergillus* species is controversial, we have previously demonstrated that treatment with 5-AC, that could inhibit the activity of DNA methyltransferase, resulted in inhibition of AF biosynthesis and developmental defects in *A. flavus* [25], which is consistent with the phenotypes defect of the disruption function of methyltransferase DmtA in *A. flavus* [28]. Intriguingly, here, we found that NT mutant produced decreasing volatile compounds compared to the wild-type strain. Our former study showed that the expression level of *veA*, a core component of the velvet complex, VeA/LaeA/VelB, was significantly downregulated in the NT mutant at a late growing stage. VeA is important for the regulation of fungal development and SMs with LaeA in *Aspergillus*, which is necessary for the activation of LaeA [29]. In *Aspergillus*, LaeA, as a global regulator of many secondary metabolisms also known as a methyltransferase, is involved in chromatin remodeling at the site of SMs gene clusters [9]. It is reasonable to see that, as the transcriptional expression level of *veA* was significantly decreased in the NT mutant, which is critical for the activation of the global regulator of SMs, LaeA, the volatile compounds produced by NT mutant are markedly reduced compared to the wild type.

Aflatoxin is the most important SMs produced by *A. flavus* which has caused severe food safety issues. Here, we found that the NT mutant was a defect in aflatoxins biosynthesis. Former studies have demonstrated that specific nutrients, like vitamins, amino acids and fatty acids are necessary for AFs formation [30]. In the AF biosynthesis pathway, it starts from acetyl-CoA, which is synthesized via oxidative decarboxylation of pyruvate, β-oxidation of fatty acids, or from ketogenic amino acids in primary metabolism [31]. In this study, we found that most of the fatty acid-derived volatiles and some of the amino acid-derived volatiles were significantly decreased in the NT mutant compared to the wild-type strain, which could explain the decreasing level of AFs in the mutant. Intriguingly, here, we found that AFB2 was detected in the hyphae of the NT strain, indicating that the secretion of AF might be impaired in the mutant. As previously reported, *A. flavus* preferentially invades oil-rich crops, which, as well as the *Aspergilli*, contain high levels of the unsaturated fatty acids, that are substrates for oxygenases [9]. These fatty acids, on one hand, are important synthesis intermediates for AF, they affect the balance between sclerotia formation and conidiation in *A. flavus* as well. The oxidative fatty acid, like oxylipins, encoded by *ppo* genes, are the signaling molecules that are involved in the regulation of developmental and SMs in *A. nidulans* [32]. Here, we found that the NT mutant failed to produce conidia, which could potentially be affected by the decreasing levels of fatty acid-derived compounds in the NT mutant.

Oxidative stress has been shown to be related to AFs biosynthesis and fungal development [33,34]. In *Aspergillus* or other filamentous fungi, the production of reactive oxygen species (ROS) and conidial germination/conidiogenesis are intimately correlated events. The mutant, which is defect in conidiation, was found to accumulate high levels of two antioxidants, including Phenol,2,2′-methylenebis[6-(1,1-dimethylethyl)-4-methyl-] and Phenol,4,4′-methylenebis[2,6-bis(1,1-dimethylethyl)-]. Our previous study demonstrated that the mutant induced by 5-AC, which showed a significant reduction in ROS accumulation, was sensitive to a strong oxidant, hydrogen peroxide. In *A. nidulans*, conidial pigmentation, mainly composed of naphthopyrones, is important for eliminating ROS for the protection of conidia against oxidative damage [35]. These data indicate that conidiogenesis might have a close relationship with oxidative stress.

## 4. Conclusions

In conclusion, we demonstrated that the NT mutant induced by the DNA methylation inhibitor, 5-azacytidine, was a severe defect in AF biosynthesis and fungal development. Additionally, we revealed that the NT mutant produced less volatile compounds, among which were the fatty acid-derived volatiles, which are important precursors for AF biosynthesis. This study may provide valuable information for controlling the contamination of mycotoxins produced by this pathogen.

## 5. Materials and Methods

### 5.1. Strains and Culture Conditions

*Aspergillus flavus* 3.4409 (A133) strain, which produces aflatoxin B1, B2, G1, and G2, was bought from the Culture Collection Center of the Chinese Academy of Sciences (Beijing, China). The NT strain is a non-aflatoxigenic mutant induced by 5-azacytidine [25]. For colony analysis, the *A. flavus* strains were cultured on PDA agar (BD Difco^TM^, Franklin Lakes, NJ, USA) at 29 °C. The Czapek liquid medium was used for the extraction of aflatoxins (AFs) and volatile metabolites. To detect the conidiophore formation, after grown on PDA medium for two days, *A. flavus* colony was observed under a light microscope.

### 5.2. Aflatoxins Analysis

For aflatoxins and volatile metabolites extraction, a total amount of 10^7^
*A. flavus* conidia was cultured into 50 mL of Czapek liquid medium at 29 °C for 5 days. After a five-day inoculation, the mycelium was harvested for the in vivo analysis of aflatoxins and volatile metabolites. Chloroform was used to extract AFs production according to the previously described method [28]. Thin Layer Chromatography (TLC) was performed to analyze AFs production.

### 5.3. GC-MS Volatile Analysis

An Agilent Model 7890A gas chromatograph (Agilent Technologies, Palo Alto, CA, USA) coupled to an Agilent 5975C Mass Selective Detector was used to detect the *Aspergillus* volatile metabolites. After being held at 160 °C for 3 min, the initial column temperature ramp at 10 °C min^−1^ to 225 °C for 3.5 min, then ramp 3 °C min^−1^ to the final temperature of 280 °C. The total run time was 70 min. The transfer line between the GC and MS systems was held at 290 °C. The ion source temperature was 230 °C.

### 5.4. Data Analysis

The resulted masses data were further analyzed by XCMS online program (https://xcmsonline.scripps.edu/) according to the published protocol [36]. The chemical structure or the chemical structural formula was analyzed in the PubChem database (https://pubchem.ncbi.nlm.nih.gov). Clustering and heat maps of the identified volatile metabolites after being analyzed with XCMS were drawn with the heatmap function in R (www.r-project.org). The top 30 volatile metabolites produced by *A. flavus* in vitro and the top 20 volatile metabolites produced by *A. flavus* in vivo that differed according to the *t*-test were further selected for analysis of heat map.

### 5.5. Statistical Analysis

In this study, the GraphPad Prism 6 was used for the analysis of statistics and significance. Student’s *t*-test was used for comparison of two different groups. One-way ANOVA multiple comparisons test was performed for significance analysis of multiple comparisons.

## Figures and Tables

**Figure 1 toxins-12-00057-f001:**
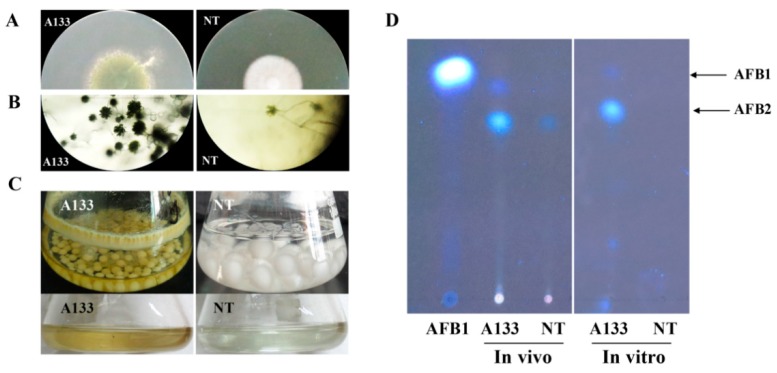
Colony morphology and aflatoxins analysis of *A*. *flavus* A133 and NT strains. (**A**) Colony morphology of *A*. *flavus* A133 and NT strains grown on PDA medium for three days. (**B**) Conidiophore formation was observed under a light microscope after 24 h incubation on PDA agar medium. (**C**) Colony morphology of *A*. *flavus* A133 and NT strains grown in Czapek liquid medium. The bottom panel was the liquid extracts of *A*. *flavus* A133 and NT strains which displayed a different color. (**D**) TLC detection of aflatoxins extracted from *A*. *flavus* A133 and NT strains in vivo and in vitro.

**Figure 2 toxins-12-00057-f002:**
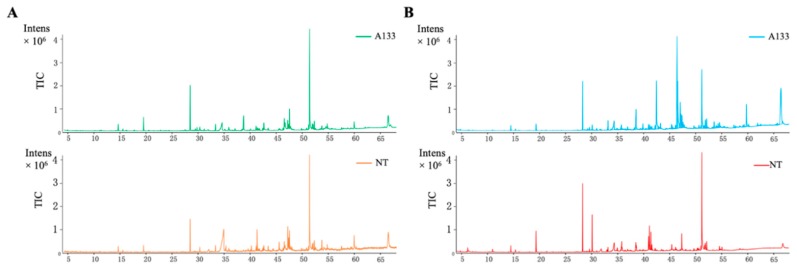
GC-MS spectra of volatiles extracted from *A*. *flavus* A133 and NT strains. (**A**) GC-MS spectra of volatiles emitted from the *A*. *flavus* A133 and NT strains (in vitro). (**B**) GC-MS spectra of volatiles extracted from the mycelium of *A*. *flavus* A133 and NT strains (in vivo).

**Figure 3 toxins-12-00057-f003:**
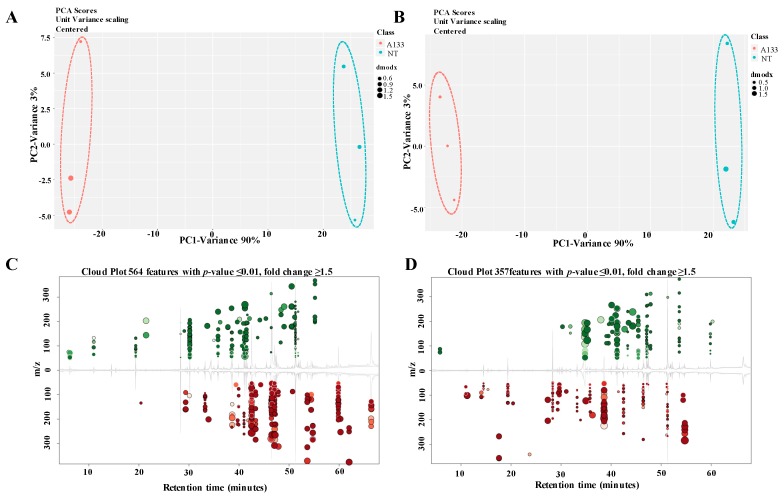
Principal components analysis (PCA) and cloud plot of the identified metabolic compounds by XCMS online program (https://xcmsonline.scripps.edu/). (**A**) Principal components analysis (PCA) of the identified in vitro volatile metabolites in WT and NT strains by XCMS online program. The PCA is calculated using the feature intensities from all samples. The colors (red/green) are assigned based on the sample class. (**B**) PCA analysis of the identified in vivo volatile metabolites in WT and NT strains by the XCMS online program. (**C**) Cloud plot of the identified in vitro volatile metabolites in WT and NT strains. (**D**) Cloud plot of the identified in vivo volatile metabolites in WT and NT strains. Only features that are dysregulated (*p*-value ≤ 0.01, fold change ≥ 1.5) are displayed. Upregulated features are shown in green, downregulated features in red. The size of each bubble corresponds to the log fold change of the feature. The shade of the bubbles corresponds to the magnitude of the *p*-value (the darker the color, the smaller the *p*-value).

**Figure 4 toxins-12-00057-f004:**
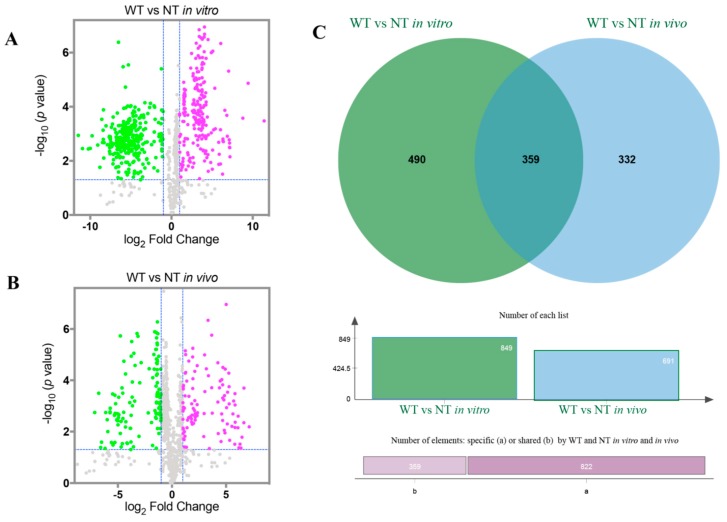
Differential volatile metabolites in the A133 (WT) strain compared to the NT strain. (**A**) Volcano plots of the differential in vitro volatile metabolites in the A133 strain compared to the NT strain. The gray spots inside the blue dash lines indicate the non-significant volatile metabolites. Volatile metabolites with *p*-value > 0.05 (−log_10_ (*p*-value) < 1.3) or log_2_ (Fold Change) < 1 are considered as non-significantly different volatile metabolites. (**B**) Volcano plots of the differential in vivo volatile metabolites in the A133 strain compared to the NT strain. The dashed line indicates *p* = 0.05. The gray spots inside the blue dash lines indicate the non-significant volatile metabolites. (**C**) The total specific in vitro volatile metabolites produced by A133 and NT strains compared to the in vivo volatile metabolites produced by these two strains.

**Figure 5 toxins-12-00057-f005:**
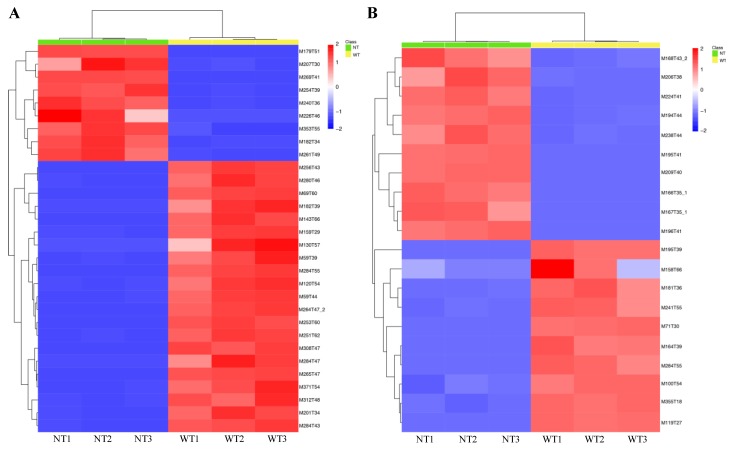
Clustering and heat map visualization of volatile metabolites in the A133 (WT) strain and NT strain in vitro and vivo. (**A**) Clustering/heat map visualization of the top 30 volatile metabolites produced by the A133 strain and NT strain in vitro differing according to *t*-test. (**B**) Clustering/heat map visualization of the top 20 volatile metabolites produced by the A133 strain and NT strain in vivo differing according to *t*-test.

**Table 1 toxins-12-00057-t001:** The top 30 volatile metabolites produced by the A133 strain and NT strain in vitro with the XCMS program. Results are derived from 3 replicates. The metabolites produced by NT mutants were compared to the A133 strain. Log_2_fold = log_2_((NT intensity)/(WT intensity)).

Name	Volatile Compound	Retention Time (min)	Quantification Ion (*m*/*z*)	Intensity		log_2_fold
				A133	NT	
Fatty acid-derived volatiles						
M256T43	n-Hexadecanoic acid	42.53	256	173,803 ± 5899	42 ± 22	−11.48
M284T43	Hexadecanoicacid,ethylester	43.31	284	10,182 ± 273	162 ± 57	−5.97
M59T44	Normeperidinic acid	44.33	205	47,057 ± 1980	711 ± 114	−6.05
M280T46	9,12-Octadecadienoicacid(Z,Z)-	46.47	280	168,927 ± 10,057	48 ± 48	−10.95
M264T47_2	6-Octadecenoic acid	46.60	264	49,278 ± 1697	0 ± 0	−9.77
M308T47	9,12-Octadecadienoicacid,ethylester	47.08	308	14,919 ± 336	0 ± 0	−8.05
M284T47	n-Propyl9-octadecenoate	47.14	284	18,212 ± 1621	0 ± 0	−8.34
M312T48	Heptadecanoicacid,15-methyl-,ethylester	47.87	312	6804 ± 330	30 ± 15	−7.12
M253T60	6,9-hexadecadienoic acid	60.28	253	42,150 ± 672	291 ± 50	−7.18
M143T66	Ergosterol	66.43	143	407,309 ± 17,088	3840 ± 72	−6.73
Phenylpropanoid derivatives/benzenoids						
M159T29	N-Methyl-N-methoxy-5,6,7,8-tetrahydro-1-naphtamide	29.33	159	73,675 ± 2920	1314 ± 197	−5.81
M207T30	Phenol,2,4-bis(1,1-dimethylethyl)-	30.16	207	636 ± 372	37,011 ± 4354	5.82
M201T34	4′-Azidobenzo[1′,2′-b]-1,4-diazabicyclo[2.2.2]octene	33.91	201	10,246 ± 446	64 ± 64	−6.66
M254T39	Phenol,3,5-dimethoxy-	38.51	254	85 ± 15	3599 ± 101	5.4
M182T39	Flopropione	38.63	182	340,108 ± 32,366	2423 ± 111	−7.13
M59T39	Phthalicacid,isobutylnonylester	39.40	59	32,073 ± 2209	818 ± 191	−5.29
M261T49	Tiaprofenic acid	49.01	260	37 ± 37	10,811 ± 627	7.18
M371T54	2,4-Difluorophenol	53.64	371	18,882 ± 1098	60 ± 60	-7.6
M353T55	Phenol,4,4′-methylenebis[2,6-bis(1,1-dimethylethyl)]-	55.18	424	1710 ± 250	16,636 ± 581	3.28
M130T57	1H-Indole,4-methyl-	57.45	130	87,806 ± 14,251	1825 ± 109	−5.59
M251T62	Propiophenone,2′-(trimethylsiloxy)-	62.00	222	40,280 ± 1350	1112 ± 147	−5.18
Triterpene						
M69T60	2,6,10,14,18,22-Tetracosahexaene,2,6,10,15,19,23-hexamethyl-,(all-E)-	59.85	410	694,833 ± 19,821	8465 ± 313	−6.36
Amino acids derived volatiles						
M182T34	L-Tyrosine	33.66	182	68 ± 34	7028 ± 315	6.34
M240T36	6-Methyl-3,5-heptadien-2-one	35.81	124	15 ± 15	4429 ± 200	6.4
M269T41	1-Hexadecanamine,N,N-dimethyl-	41.18	269	0 ± 0	39,265 ± 145	9.44
M226T46	N-Methyl-beta-carboline-3-carboxamide	46.1	225	0 ± 0	4410 ± 794	6.29
M265T47	9-Octadecenamide,(Z)-	47.25	265	18,661 ± 213	0 ± 0	−8.37
M179T51	2-phenyl-N-(5-propan-2-yl-1,3-thiazol-2-yl)acetamide	50.52	260	1332 ± 173	89,236 ± 370	6.07
M120T54	4-Morpholineethanol	54.40	131	57,950 ± 3565	562 ± 108	−6.69
M284T55	1-(2-Thiazolylazo)-2-naphthol	54.74	255	33,915 ± 1238	170 ± 25	−7.64

**Table 2 toxins-12-00057-t002:** The top 20 volatile metabolites produced by the A133 strain and NT strain in vivo with the XCMS program. Results are derived from 3 replicates. The metabolites produced by the NT mutants were compared to the A133 strain. Log_2_fold = log_2_((NT intensity)/(WT intensity)).

Name	Volatile Compound	Retention Time (min)	Quantification Ion (*m*/*z*)	Intensity		log_2_fold
				A133	NT	
Fatty acid-derived volatiles						
M119T27	Trimethylammonioacetate	27.35	117.15	32,031 ± 246	1471 ± 71	−4.44
M100T54	Hexadecanoicacid,2-hydroxy-1-(hydroxymethyl)ethylester	54.40	330.5	36,178 ± 753	3406 ± 1018	−3.41
M284T55	1,2-Benzenedicarboxylicacid,mono(2-ethylhexyl)ester	54.74	54.74	53,296 ± 2260	378 ± 61	−7.14
M158T66	Ergosterol	66.44	143	16,097 ± 6460	897 ± 897	−3.47
Phenylpropanoid derivatives/benzenoids						
M355T18	2,6-Dichloroindophenol	17.59	268.1	33,856 ± 310	2579 ± 469	−3.71
M167T35_1	Benzo[b]tetrahydrofuran-3-one,5,6-dihydroxy-	34.79	166.027	1110 ± 35	111,478 ± 7417	6.65
M166T35_1	2,6-toluenediamine	35.23	122.17	4043 ± 615	211,094 ± 5606	5.71
M209T40	Isoelemicin	40.11	208.25	13,703 ± 254	353,432 ± 4258	4.69
Saccharide derived volatiles						
M195T41	D-Glucoside	41.2	194.18	34,031 ± 185	1,301,860 ± 9210	5.26
Amino acids derived volatiles						
M71T30	Caulophylline	29.66	204.27	112,547 ± 868	12,794 ± 455	−3.14
M181T36	Theobromine	36.27	180.16	11,780 ± 637	382 ± 66	−4.95
M206T38	N-Acetyl-D-quinovosamine	37.95	205.21	170 ± 64	7878 ± 634	5.54
M195T39	N,N′-Diacetyl-2-nitro-p-phenylenediamine	38.6	237.21	574,864 ± 10,207	5272 ± 227	−6.77
M164T39	2,2′-(2-Hydroxy-2-nitrosohydrazinylidene)bis-ethanamine	38.61	163.17	54,132 ± 2206	1027 ± 49	−5.72
M224T41	4-(Methylnitrosamino)-1-(1-oxido-3-pyridinyl)-1-butanone	40.90	223.23	293 ± 38	7571 ± 196	4.69
M196T41	9-Aminoacridine	41.19	194.23	4102 ± 160	145,064 ± 2468	5.14
M168T43_2	2,8-Dihydroxyadenine	42.71	167.12	455 ± 174	11,735 ± 805	4.69
M194T44	Phenylacetylglycine	43.50	194.1	1457 ± 103	18,559 ± 371	3.67
M238T44	N-Benzylphthalimide	44.31	237.25	469 ± 201	32,530 ± 1702	6.12
M241T55	Tetramisole hydrochloride	54.74	240.75	9247 ± 476	243 ± 101	−5.25

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
