# Peer review of "Gas Chromatography–Mass Spectrometry Profiling of Volatile Compounds Reveals Metabolic Changes in a Non-Aflatoxigenic Aspergillus flavus Induced by 5-Azacytidine"

_toxins, 2020, doi:10.3390/toxins12010057_

Round 1
Reviewer 1 Report
The manuscript under review measures the VOCs secreted by two strains of A. flavus to study changes due to mutations that reduce aflatoxin production by the fungus. The methods presented use well-documented tools which should provide reliable results. In general, the work appears sound and deserves publication after minor revision to the presentation and figure improvements.
Strange choice in figure 2. The jaunty angle of the data presented makes comparing the data directly difficult. Please plot these data on simple x:y graphs, offset for clarity.
In all the figures, the labels and captions are far too small to read. The PCA figure in particular is impossible to understand. There are ovals that are drawn roughly around some scattered plots. I assume that this is some type of visual grouping along PC axis 1, but there is no attempt to describe this in the text. The caption provides no help and the legend is illegible.
In general, the data and figures need to be described more completely. It is not sufficient to simply reference the figure. This is most glaring in the instance the discussion around line 147, “These top differential expressed metabolites were classified, which included fatty, amino, organic acid, as well as the saccharides.” This may well be true, but nothing in the text or figures prior to this statement suggests that this analysis was done thoroughly. My suspicion is that this is a problem of organization of the manuscript rather than a faulty analysis.
Author Response
Comments and Suggestions for Authors
The manuscript under review measures the VOCs secreted by two strains of A. flavus to study changes due to mutations that reduce aflatoxin production by the fungus. The methods presented use well-documented tools which should provide reliable results. In general, the work appears sound and deserves publication after minor revision to the presentation and figure improvements.
1. Strange choice in figure 2. The jaunty angle of the data presented makes comparing the data directly difficult. Please plot these data on simple x:y graphs, offset for clarity.
Answer:Thank you for your good suggestion, wehave revised the figure 2 according to your suggestion. (line 122).
2. In all the figures, the labels and captions are far too small to read.
Answer:Thank you for your good suggestion, we had provided a higher resolutions of figures in the manuscript (line122, 133).
3. The PCA figure in particular is impossible to understand. There are ovals that are drawn roughly around some scattered plots. I assume that this is some type of visual grouping along PC axis 1, but there is no attempt to describe this in the text. The caption provides no help and the legend is illegible.
Answer:Thank you for your good suggestion, we had re-plotted the PCA figure. We had reinterpreted the PCA figure. Metabolite features that drive this sample clustering can be identified on the loadings plot. The metabolite features that show the largest possible variance lie on the first principal component (PC1) and those that show subsequent largest variance lie on the second principal component (PC2). (line 117-120, line 136-138).
4. In general, the data and figures need to be described more completely. It is not sufficient to simply reference the figure. This is most glaring in the instance the discussion around line 147, “These top differential expressed metabolites were classified, which included fatty, amino, organic acid, as well as the saccharides.” This may well be true, but nothing in the text or figures prior to this statement suggests that this analysis was done thoroughly. My suspicion is that this is a problem of organization of the manuscript rather than a faulty analysis.
Answer:Thank you for your good suggestion, we had reinterpreted most of results as your suggestions (line 97-98, 234-238, 256-258, 273-277, 299-301).
Reviewer 2 Report
Manuscript Recommendation: Accept
The manuscript titled “Volatile profiling reveals metabolic changes in a non-aflatoxigenic Aspergillus flavus induced by DNA methylation inhibitor using gas chromatography–mass spectrometry” investigates the metabolic changes between two strains of Aspergillus flavus. A total of 1181 volatiles were identified in two strains by the authors, among which 490 volatiles were found in these strains in vitro and 332 volatiles were found in vivo. These results reveal the difference of metabolic profiles in two different A. flavus isolates, which may provide valuable information for controlling infections of this fungal pathogen. This is an important field of study given the overall burden of this fungal pathogen. The results support the overall conclusions of the study. I would recommend accepting the manuscript in its current form.
Author Response
Thank you for so much for your positive comments.
Reviewer 3 Report
In the research paper entitled “Volatile profiling reveals metabolic changes in a non-aflatoxigenic Aspergillus flavus induced by DNA methylation inhibitor using gas chromatography-mass spectrometry”, the authors compared volatile compounds produced by either WT A. flavus or mutant that is defective for full production of aflatoxin. Although the findings have merits for readers in foodborne toxin fields, following concerns should be revisited before considering publication in the Toxins journal.
Major concerns
Authors arguing that this study may provide valuable information for infection controls. But how can the results - mostly just metabolite profiles- contribute for controlling fungal infections? They state that two antioxidant molecules are overproduced in the mutant than the wildtype. I wonder that the molecules actually work as antioxidants in the cells, or in the physiological conditions. Any references showing the biological roles of the molecules should be included at least. They state that most of the fatty acid derived volatile molecules and some of the amino acid derivatives were significantly decreased in the mutant, and this could explain the decreasing level of AF in the mutant. Are these decreased volatile compounds directly related with the AF production? Please present evidences or references. The biosynthetic pathway for AF production in A. flavus may also be helpful for the readers. What are the genetical reasons for the NT mutant? (what kinds of genes are mutated in the NT mutant?) Have this ever been studied? Discussion regarding that information need to be included. What is the difference between XCMS online program and Agilent ChemStation software. Why the identified metabolite lists are so different?
Minor concerns
In the entire manuscript (even in the title), ‘volatile’ was used as noun. This should be corrected as volatile metabolite(s) or volatile compound(s). Title may be revised. GC-MS based profiling of volatile compounds reveals… line 35. ->The whole genome of A. flavus has been sequenced. Not only this but many other sentences need revision. For example, line 36: -> Although each SM cluster can ....., only AFs... (you can remove 'however, just mentioned above') line 38: identified as what? line 42-43 can be removed. Antibiotic resistance is not the subject of this study. line 98: To assayed-> To analyze the differences Figure legends and related main text for Fig. 3 and 4: Need extensive improvements in English. line 113. -> Principal components analysis (PCA) plots and cloud plots of the identified metabolic compounds. The following sentences also need to be revised. Figure 3, legend: about cloud plots, details should be explained. For example, the meanings of the color, size of circles, etc. line 121, also in figure: 10 and 2 should be subscript. Figure 4C, Do the numbers mean total volatile metabolites? Or the statistically different volatile metabolites? line 130: non-significantly different volatile metabolites..? line 135. Sub-heading should be revised as it is almost same with the subheading of 2.2. Table 1. Explanation sentence need revision (English). And this should locate under the table. Table 1 and 2. What does number in A133 and NT strain columns mean? Is the relative amount? What is the unit for that? How the ‘log2 fold’ calculated? This kind of information should be included in the table legends. Even in the tables in supple should be revised. line 201. What is the biological meaning of the reduced volatile compounds in the NT mutant? Not just results but discussion should be presented in the discussion section. line 203~213. Redundant with the introduction section. line 225. In figure 1, NT actually produces detectable amount of AF B2 in vivo, although it’s amount was reduced. Therefore, the sentence in line 225 is not correct.Author Response
Comments and Suggestions for Authors
In the research paper entitled “Volatile profiling reveals metabolic changes in a non-aflatoxigenic Aspergillus flavus induced by DNA methylation inhibitor using gas chromatography-mass spectrometry”, the authors compared volatile compounds produced by either WT A. flavus or mutant that is defective for full production of aflatoxin. Although the findings have merits for readers in foodborne toxin fields, following concerns should be revisited before considering publication in the Toxins journal.
Major concerns
Authors arguing that this study may provide valuable information for infection controls. But how can the results - mostly just metabolite profiles- contribute for controlling fungal infections?Answer: Thank you for so much for your good question,in this study we used the DNA methylation inhibitor to treat A. flavus, and found that A. flavuswas decrease in AF biosynthesis and impaired in fungal development. The metabolite profiles might account for the changes in the SMs in this fungus. To avoid misunderstand the readers, we have revised this in the text (line 324).
They state that two antioxidant moleculesare overproduced in the mutant than the wildtype. I wonder that the molecules actually work as antioxidants in the cells, or in the physiological conditions. Any references showing the biological roles of the molecules should be included at least.Answer:Thank you for so much for your good suggestions,the properties of the metabolites were identified both in the websites of pubchem database (https://pubchem.ncbi.nlm.nih.gov) and Guidechem (https://www.guidechem .com/). It would be interesting to investigate the roles of these two antioxidant molecules in A. flavus in our further work. (line 234-238)
They state that most of the fatty acid derived volatile molecules and some of the amino acid derivatives were significantly decreased in the mutant, and this could explain the decreasing level of AF in the mutant. Are these decreased volatile compounds directly related with the AF production? Please present evidences or references. The biosynthetic pathway for AF production in A. flavus may also be helpful for the readers.Answer:Thank you for so much for your good suggestions, they are not directly related with AF production. AF biosynthesis pathway starts from acetyl-CoA, which is synthesized via oxidative decarboxylation of pyruvate, β-oxidation of fatty acids, or from ketogenic amino acids in primary metabolism.We have indicated the detail in the discussion section (line 275-277)
What are the genetical reasons for the NT mutant? (what kinds of genes are mutated in the NT mutant?) Have this ever been studied? Discussion regarding that information need to be included.Answer:Thank you for so much for your good question,the NT mutant was resulted from the treatment of the DNA methylation inhibitor 5-azacytidine, which has been reported by our former study (Yang et al 2015). One of our further work is going to use the de novo DNA sequencing to know its genetical mutation (line 248).
What is the difference betweenXCMS online programand Agilent ChemStation software. Why the identified metabolite lists are so different?Answer:Thank you for so much for your good question,Agilent ChemStation software is based on Nist chemical database, which could provide quality for each metabolites alignment with the Nist database. XCMS online program was originally developed as a metabolomics data processing algorithm to extract metabolic features from raw MS data and perform statistical analysis, which is based on exact Mass algorithm. To screen much more volatile compounds inA. flavus, both of this two methods were used in this study.
Minor concerns
In the entire manuscript (even in the title), ‘volatile’ was used as noun. This should be corrected as volatile metabolite(s) or volatile compound(s). Title may be revised. GC-MS based profiling of volatile compounds reveals…Answer:Thank you for so much for your suggestions, we have revised it as red words (line 2-5).
line 35. ->The whole genome of A. flavushas been sequenced.Answer:Thank you for so much for your suggestions, we have revised it as red words (line 35).
Not only this but many other sentences need revision. For example, line 36: -> Although eachSM cluster can ....., only AFs... (you can remove 'however, just mentioned above')Answer: Thank you for so much for your suggestions, we have revised it as red words (line 36-37).
line 38: identified as what?Answer: Thank you for your good question, here we meant to identify the molecular structures of these SMs, and we have revised it as red words (line 37).
line 42-43 can be removed. Antibiotic resistance is not the subject of this study.Answer: Thank you for your good suggestions, we have removed this sentence according to your suggestion (line 42).
line 98: To assayed-> To analyze the differences Figure legends and related main text for Fig. 3 and 4: Need extensive improvements in English.Answer:Thank you for your good suggestions, we have revised it according to your suggestion (line 111, 117-120, 141-144).
line 113. -> Principal components analysis (PCA) plots and cloud plots of the identified metabolic compounds. The following sentences also need to be revised. Figure 3, legend: about cloud plots, details should be explained. For example, the meanings of the color, size of circles, etc.Answer: Thank you for your good suggestions, we have added more details in the figure legend (line 137-144).
line 121, also in figure: 10 and 2 should be subscript.Answer:Thank you for your good suggestions, we have revised these in the text and figure (line 148, 154).
Figure 4C, Dothe numbers mean total volatile metabolites? Or the statistically differentvolatile metabolites?Answer: Thank you for so much for your good questions, the numbers mean the numbers mean total volatile metabolites volatile metabolites, which are also thestatistically different metabolites.
line 130: non-significantly different volatile metabolites..?Answer: Thank you for so much for your good questions,volatile metabolites with p value >0.05 (-log10 p-value<1.3) or log2 Fold Change < 1 are considered as non-significantly different volatile metabolites(line157-159).
line 135. Sub-heading should be revised as it is almost same with the subheading of 2.2.Answer:Thank you for your good suggestions, we have revised the sub-heading of 2.4 (line 164).
Table 1. Explanation sentence need revision (English). And this should locate under the table. Table 1 and 2. What does number in A133 and NT strain columns mean? Is the relative amount? What is the unit for that?Answer:Thank you for so much for your good questions,we have revised these according to your suggestions. The number in A133 and NT strain columns meanthe intensity of the volatile metabolites identified in the XCMS programme (line 211-213, 216-218).
How the ‘log2 fold’ calculated? This kind of information should be included in the table legends. Even in the tables in supple should be revised.Answer:Thank you for your good suggestions, log2fold is equal to log2((NT intensity)/ (WT intensity)), we have added the information in the text (line 213, 218).
line 201. What is the biological meaning of the reduced volatile compounds in the NT mutant? Not just results but discussion should be presented in the discussion section.Answer:Thank you for your good suggestions, we think that the decreasing volatile compounds observed in the NT mutant, which could result in such physiological changes as reducing AF production and asporulate phenotype in NT mutant. (line 251-252).
line 203~213. Redundant with the introduction section.Answer: Thank you for your good suggestions, we have revised this according to your suggestion (line 254-255).
line 225. In figure 1, NT actually produces detectable amount of AFB2 in vivo, although it’s amount was reduced. Therefore, the sentence in line 225 is not correct.Answer: Thank you for your good suggestions, we have revised it as red word (line 273, 299-301).

Reviewer 4 Report
This is a review of “Volatile profiling reveals metabolic changes in a non-aflatoxigenic Aspergillus flavus induced by DNA methylation inhibitor using gas chromatography mass spectrometry.” In this manuscript, the authors characterize volatile compounds produced by wild-type and a non-aflatoxigenic strains of Aspergillus flavus. The authors draw a correlation between the accumulation of two antioxidants and the downregulation of aflatoxin biosynthesis. Although the methodology seems basically valid and the results seem interesting, there are numerous issues in the manuscript that need clarification. This author recommends major modifications.
Major Points:
1). The genetic stability and basis of the NT mutant is poorly described, and considering that the role of DNA methylation in gene expression in Aspergillus is still controversial, the authors need to describe the stability and molecular basis of the NT phenotype in greater detail.
2). The difference between “in vivo” vs “in vitro,” as presented in Figure 1 is not defined. What exactly are “in vivo” conditions and why is AFB2 detected in the mutant under “in vivo” conditions (Figure 1D)?
3). Data given in Table 1 and 2 need better definition. What are the units in the fourth column and fifth column under A133 and NT? Are five figure data points truly significant?
Minor points:
1). There are numerous minor grammatical errors and wording issues throughout the manuscript. Attention should be paid to the wording and spelling on lines 9, 23, 31, 34, 35, 42, 88, 240, and 251. However, there are likely other lines that need to be corrected. Instead of pointing each of these, my suggestion would be to have a language editor carefully review the manuscript for phrasing.
2). The authors should shorten the title.
3). The authors need to review carefully that the abbreviations are defined before being used in the text or abstract. GS-MS should be defined.

Author Response
Comments and Suggestions for Authors
This is a review of “Volatile profiling reveals metabolic changes in a non-aflatoxigenic Aspergillus flavus induced by DNA methylation inhibitor using gas chromatography mass spectrometry.” In this manuscript, the authors characterize volatile compounds produced by wild-type and a non-aflatoxigenic strains of Aspergillus flavus. The authors draw a correlation between the accumulation of two antioxidants and the downregulation of aflatoxin biosynthesis. Although the methodology seems basically valid and the results seem interesting, there are numerous issues in the manuscript that need clarification. This author recommends major modifications.
Major Points:
1). The genetic stability and basis of the NT mutant is poorly described, and considering that the role of DNA methylation in gene expressionin Aspergillus is still controversial, the authors need to describe the stability and molecular basis of the NT phenotype in greater detail.
Answer:Thank you for so much for your good questions, although the role of DNA methylation in gene expression in Aspergillus is still controversial, it’s interesting to see that the DNA methylation inhibitor 5-azacytidine resulted in decreased AF production and concurrent morphological changes in A. flavus. Former investigations have shown that no AF production was detected in 5-AC-treated A. parasiticus clones as well. Our former study (Yang et al 2015) has studied the hereditary stability of this mutant, and indicated that the phenotype defects are hereditarily stable. 5-azacytosine, as a derivative of the nucleosidecytidine, it can be served as a DNA methyltransferase inhibitor,and might be served as mutagen for the substitution of nucleoside (cytidine) in the treatment genome.One of our further work is going to use the de novo DNA sequencing to know its genetical mutation.
2). The difference between “in vivo” vs “in vitro,” as presented in Figure 1 is not defined. What exactly are “in vivo” conditions and why is AFB2 detectedin the mutant under “in vivo” conditions(Figure 1D)?
Answer:Thank you for so much for your suggestions, metabolites in vivo are extracted from the mycelium of A. flavus,and metabolites in vitroare volatiles emitted from the A. flavus, we have indicated the details in the text (line 97-98). Although AF could not be detected in the in the broth cultures (in vitro), AFB2 was detected in the mutant under “in vivo” conditions, which indicated that the secretion of AF might be impaired in this mutant (line 299-301).
3). Data given in Table 1 and 2 need better definition. What are the units in the fourth column and fifth column under A133 and NT? Are five figure data points truly significant?
Answer:Thank you for so much for your suggestions,we have revised these according to your suggestions. The data of fourth column and fifth column under A133 and NT indicates the intensity of each metabolite feature (line 202, 215). The figure 5 indicates the top significantly different metabolites in these two strains, and the detail information of these metabolites were displayed in table 1 and table 2.
Minor points:
1). There are numerous minor grammatical errors and wording issues throughout the manuscript. Attention should be paid to the wording and spelling on lines 9, 23, 31, 34, 35, 42, 88, 240, and 251. However, there are likely other lines that need to be corrected. Instead of pointing each of these, my suggestion would be to have a language editor carefully review the manuscript for phrasing.
Answer:Thank you for so much for your suggestions, we have revised the spelling errors according to your suggestions.
2). The authors should shorten the title.
Answer:Thank you for so much for your suggestions, we have revised the title according to your suggestions and other reviewer’s comments (line 2-5).
3). The authors need to review carefully that the abbreviations are defined before being used in the text or abstract. GS-MS should be defined.
Answer:Thank you for so much for your suggestions, the abbreviations are newly defined in the text (line 10-11).
Round 2
Reviewer 3 Report
Thank you for appropriate revision. I hope the comments help improve the manuscript.
Author Response
Comments and Suggestions for Authors
Thank you for appropriate revision. I hope the comments help improve the manuscript.
Answer:Thank you for so much for your positive comments.
Reviewer 4 Report
The authors have addressed my concern. Please check, however, for minor grammatical and spelling errors. I urge the reviewers to carefully consider the significant digits in the Tables.
Author Response
Comments and Suggestions for Authors
The authors have addressed my concern. Please check, however, for minor grammatical and spelling errors. I urge the reviewers to carefully consider the significant digits in the Tables.
Answer: Thank you for so much for your good suggestions, we found your comments were very helpful to this manuscript, and we have carefully revised the manuscript according to your suggestions.
For the significant digits in the Tables, we found that the editor and other reviewers haven’t addressed this issue, so we didn’t edit the tables.